# *Salvia sclarea* Essential Oil Chemical Composition and Biological Activities

**DOI:** 10.3390/ijms24065179

**Published:** 2023-03-08

**Authors:** Miroslava Kačániová, Nenad L. Vukovic, Natália Čmiková, Lucia Galovičová, Marianna Schwarzová, Veronika Šimora, Przemysław Łukasz Kowalczewski, Maciej Ireneusz Kluz, Czeslaw Puchalski, Ladislav Bakay, Milena D. Vukic

**Affiliations:** 1Institute of Horticulture, Faculty of Horticulture and Landscape Engineering, Slovak University of Agriculture, Tr. A. Hlinku 2, 94976 Nitra, Slovakia; 2Department of Bioenergetics and Food Analysis, Institute of Food Technology and Nutrition, University of Rzeszow, Zelwerowicza 4, 35-601 Rzeszow, Poland; 3Department of Chemistry, Faculty of Science, University of Kragujevac, 34000 Kragujevac, Serbia; 4Department of Food Technology of Plant Origin, Poznań University of Life Sciences, 31 Wojska Polskiego St., 60-624 Poznań, Poland; 5Institute of Landscape Architecture, Faculty of Horticulture and Landscape Engineering, Slovak University of Agriculture, Tr. A. Hlinku 2, 94976 Nitra, Slovakia

**Keywords:** *Salvia sclarea*, (*E*)-caryophyllene, antimicrobial activity, antibiofilm activity, microorganisms, vapor phase

## Abstract

*Salvia sclarea* essential oil (SSEO) has a long tradition in the food, cosmetic, and perfume industries. The present study aimed to analyze the chemical composition of SSEO, its antioxidant activity, antimicrobial activity in vitro and in situ, antibiofilm, and insecticidal activity. Besides that, in this study, we have evaluated the antimicrobial activity of SSEO constituent (*E*)-caryophyllene and standard antibiotic meropenem. Identification of volatile constituents was performed by using gas chromatography (GC) and gas chromatography/mass spectrometry (GC/MS) techniques. Results obtained indicated that the main constituents of SSEO were linalool acetate (49.1%) and linalool (20.6%), followed by (*E*)-caryophyllene (5.1%), *p*-cimene (4.9%), a-terpineol (4.9%), and geranyl acetate (4.4%). Antioxidant activity was determined as low by the means of neutralization of the DDPH radical and ABTS radical cation. The SSEO was able to neutralize the DPPH radical to an extent of 11.76 ± 1.34%, while its ability to decolorize the ABTS radical cation was determined at 29.70 ± 1.45%. Preliminary results of antimicrobial activity were obtained with the disc diffusion method, while further results were obtained by broth microdilution and the vapor phase method. Overall, the results of antimicrobial testing of SSEO, (*E*)-caryophyllene, and meropenem, were moderate. However, the lowest MIC values, determined in the range of 0.22–0.75 µg/mL for MIC50 and 0.39–0.89 µg/mL for MIC90, were observed for (*E*)-caryophyllene. The antimicrobial activity of the vapor phase of SSEO (towards microorganisms growing on potato) was significantly stronger than that of the contact application. Biofilm analysis using the MALDI TOF MS Biotyper showed changes in the protein profile of *Pseudomonas fluorescens* that showed the efficiency of SSEO in inhibiting biofilm formation on stainless-steel and plastic surfaces. The insecticidal potential of SSEO against *Oxycarenus lavatera* was also demonstrated, and results show that the highest concentration was the most effective, showing insecticidal activity of 66.66%. The results obtained in this study indicate the potential application of SSEO as a biofilm control agent, in the shelf-life extension and storage of potatoes, and as an insecticidal agent.

## 1. Introduction

Essential oils (EOs) represent oily liquids that consist of a complex mixture of volatile secondary metabolites. From a chemical point of view, they mainly belong to the class of terpenoids with characteristic flavor and fragrance properties [1,2]. These volatile liquids are produced in aromatic plants, where they have a role in attracting insects and other pollinators crucial to the life of a plant. Moreover, EOs can be involved in plant protection, i.e., from different herbivores and pathogens, as well as from environmental stress. Besides all the benefits they provide to plants, EOs have been found to have useful applications in everyday human life, too. So far, they have played an important role in food protection, acting as antibacterials, antivirals, antifungals, and antioxidants [3]. Nowadays, essential oils are widely used, especially because they are recognized as safe substances by the Food and Drug Administration and the Environmental Protection Agency, as well as for containing compounds that can be used as antibacterial additives [4,5,6]. 

*Salvia sclarea* L., commonly known as clary sage, belongs to the Lamiaceae family. This biennial or perennial plant species is native to southern Europe, but it is cultivated worldwide as an ornamental and essential oil (EO)-bearing plant, mainly in regions with temperate and sub-tropical climates [7,8,9,10]. The *S. sclarea* plants can reach a height of up to 130 cm. Their flowering spikes can reach up to 40 cm in length, and their lilac to whitish cymose inflorescence is characterized by an assemblage of axillary flowers in clusters that are subtended by bracts [8,11,12]. Since ancient times, EO obtained from *Salvia* species has been used due to its aromatic and medicinal properties, but nowadays it has a significant economic value in the perfume and cosmetics industries [13]. Traditionally, these species have been used as agents against inflammatory conditions (gingivitis, stomatitis, and aphthae), for the common cold and nervous fatigue, to increase lactation, and to prevent extreme sweating [8,13,14]. Reports made until now show that these species display a broad spectrum of biological activities such as antibacterial, anti-inflammatory, antifungal, antiviral, antiseptic, analgesic, antioxidant, anticancer, and antidiabetic [1,7,9,13,14].

*S. sclarea* EO (SSEO) mainly consists of different types of terpene compounds such as monoterpene and sesquiterpene hydrocarbons, oxygenated monoterpenes and sesquiterpenes, aliphatic alcohols, and esters [13]. Among them, the most commonly found compounds are linalool and linalyl acetate as main compounds, followed by high amounts of α-terpineol, 1,8-cineole, sclareol, germacrene D, caryophyllene, and caryophyllene oxide [8,10,14,15,16]. These volatile compounds are found to be responsible for the reported beneficial effects of SSEO. Commercially, this EO is used in the food industry as a flavoring agent for different desserts, beverages, and pastries [8,9]. Moreover, it is used as an aromatic agent in tobacco and as a scent component in different cosmetic formulas and perfumes [9,10]. Due to its antimicrobial properties, it can also be used to prevent food spoilage [2].

One of the plant’s secondary metabolites groups that is widely spread in nature belongs to the class of sesquiterpenoids. The carbon backbone of these compounds consists of 15 atoms, and their general formula is C_15_H_24_ or C_15_H_32_ [17,18]. Among them, the caryophyllene group of bicyclic sesquiterpenes represents the smallest and most widely distributed group in nature. Naturally, β-caryophyllene occurs mainly as its trans isomer, ((*E*)-caryophyllene), but it also can be found in the form of its isomers, (Z)-caryophyllene (iso-caryophyllene) and α-caryophyllene (α-humulene), as well as its oxidation derivative, β-caryophyllene oxide [19]. It can be mainly found in the essential oil extracts of plants as its major active compound, and its use as a flavoring agent is approved by the Food and Drug Administration (FDA) and the European Food Safety Authority (EFSA) [19]. The biological effects of this sesquiterpenoid define it as an anti-inflammatory, anticarcinogenic, antimicrobial, antioxidant, and analgesic agent [17,19,20,21]. Moreover, a pre-clinical study reported that β-caryophyllene, as a modulator of the nervous system, shows favorable effects on numerous neurodegenerative and inflammatory pathologies [17].

Meropenem is a parenteral antibiotic from the carbapenems family. This broad-spectrum antibiotic has been in clinical use since the 1990s. Reports made until now show the activity of meropenem towards both G^−^ and G^+^ bacterial strains [22,23,24,25,26]. Now, this is the second-most-used antibiotic because of its broader antibacterial coverage as well as its high potential in monotherapy as an alternative to combination therapy [26]. It has been mainly used to treat severe infections, such as sepsis and meningitis [24]. The mechanism of action of this antibiotic consists of the prevention of bacterial cell wall synthesis by binding and inactivating penicillin-binding proteins and by inhibiting the cross-linking of peptidoglycan chains [24]. However, because of its low solubility, this antibiotic is available only as a parenteral formula administered by continuous infusion or bolus injection [24,25,26]. Moreover, using meropenem causes adverse events such as diarrhea, nausea, hypersensitivity reactions, seizures, and other central nervous system disorders, etc. [24]. Recently, there has been increasing concern that meropenem exposure might be inadequate, particularly in the treatment of critically ill patients, with the requirement for individualized treatment [26]. Likewise, multi-antibiotic drug-resistant G^−^ bacilli has become a major concern in standard patient care [27].

Bearing in mind the abovementioned, the main goal of this investigation was to define the volatile compounds of *Salvia sclarea* essential oil (SSEO) using the GC-MS technique, assess its antioxidant activity, in vitro and in situ antimicrobial activity, as well as determine its antibiofilm and insecticidal activity. Moreover, the antimicrobial activity of the standard antibiotic meropenem and (*E*)-caryophyllene, one of the constituents of this EO, was determined. 

## 2. Results

### 2.1. Chemical Composition

Results of the chemical composition of SSEO are obtained using GC and GC/MS analysis and are presented in Table 1 and Table 2. The percentage amount of each volatile identified is presented in Table 1, while Table 2 shows the percentage composition of the class of compounds. Generally, in the *S. sclarea* examined in this study, thirty-nine compounds were identified, which represent 99.6% of the total. The major compound identified was monoterpene ester linalool acetate (49.1%), followed by monoterpene alcohol linalool (20.6%). Both major compounds belong to the class of oxygenated monoterpenes, represented by an overall contribution of 82.4%. In high amounts were also detected sesquiterpene hydrocarbon (*E*)-caryophyllene (5.1%), monoterpene hydrocarbon *p*-cymene (4.9%), and monoterpene ester geranyl acetate (4.4%). The *S. sclarea* EO tested in this study has also been characterized by a considerable quantity of α-pinene (2.4%), α-limonene (2.2%), and neryl acetate (1.7%), while the rest of the identified compounds were found in amounts less than 1.5%.

### 2.2. Antioxidant Activity

Antioxidant activity of *S. sclarea* EO was measured using DPPH and ABTS assays. DPPH radical scavenging ability was determined at 11.76 ± 1.34% inhibition, which is equivalent to 1.37 ± 0.10 TEAC, while the IC50 value for Trolox standard was determined at 4.39 ± 0.13 μg/mL. Results obtained by ABTS assay show that SSEO decolorizes the ABTS radical cation to the extent of 29.70 ± 1.45% which is equivalent to 1.99 ± 0.04 TEAC, while the IC50 value for the Trolox standard was determined at 2.96 ± 0.01 μg/mL. These results indicate that the overall antioxidant activity of the SSEO is lower compared to the tested standard Trolox. 

### 2.3. Antimicrobial Activity In Vitro 

#### 2.3.1. Disc Diffusion Method

The antimicrobial effects of *S. sclarea* EO (SSEO) and its constituent (*E*)-caryophyllene were first determined by using the disc diffusion method, and the obtained results are presented in Table 3 and Table 4, respectively. 

The antimicrobial effects of *S. sclarea* EO generally indicate moderate activity towards tested microbials. Strong effects of SSEO were noted for the G^+^ bacterial strain *B. subtilis* (12 ± 1.00 mm) and the yeast *C. albicans* (11.33 ± 0.58 mm). Considering fungi strains, the strongest activity of tested EO was observed towards *A. flavus* with an inhibition zone of 10.33 ± 0.58 mm. The highest resistance on the exposure with SSEO was observed for G^−^
*Y. enterocolitica* and *S. enterica*, with an inhibition zone determined at 3.67 ± 0.58 mm for both, and G^+^
*E. faecalis* with an inhibition zone of 4.67 ± 0.58 mm. Treatment with SSEO showed moderate activity on inhibition of other microorganisms, with inhibition zones ranging from 6.67 ± 0.58 mm to 9.67 ± 0.58 mm. 

The effects of (*E*)-caryophyllene on different microbial strains are presented in Table 4. Overall, obtained results indicate moderate activity of (*E*)-caryophyllene towards all tested strains except G^+^
*B. subtilis*, where only a weak effect was observed with an inhibition zone of 3.67 ± 0.58 mm. The best antimicrobial activity of this compound was detected against the yeast *C. krusei* and G^−^ bacterial strain *Y. enterocolitica* with inhibition zones of 7.67 ± 0.58 mm and 6.67 ± 0.58 mm, respectively. Towards all other microbial species, (*E*)-caryophyllene showed moderate activity with an inhibition zone of 5.33 ± 0.58 mm and 5.67 ± 0.58 mm.

Standard antibiotics meropenem and fluconazole, tested as a positive control, showed considerably strong activity towards tested microorganisms that were in the range of 27 ± 1.5 to 35 ± 0.5 mm. 

In general, it can be concluded that SSEO showed better antimicrobial effects compared to (*E*)-caryophyllene. However, compared to the representative controls, meropenem and fluconazole, both SSEO and (*E*)-caryophyllene showed considerably weaker effects. 

#### 2.3.2. Minimal Inhibitory Concentration Assay

In order to further evaluate the antimicrobial effects of SSEO and (*E*)-caryophyllene, we have used a minimal inhibitory concentration assay. The results of this study are presented in Table 5 and Table 6. 

The effects of SSEO obtained by this assay indicate the best antimicrobial activity of SSEO against G^+^ against *S. aureus* (MIC50 of 1.48 and MIC90 of 1.59 µL/mL). Out of the tested G^−^ bacteria, the best antimicrobial activity was found against biofilm-forming *P. flourescens* with a MIC50 of 2.93 and a MIC90 of 3.17 µL/mL. Considering the yeasts, the best effect of SSEO was detected against *C. tropicalis* (MIC50 of 2.93 and MIC90 of 3.17 µL/mL). The most resistant to the treatment with this essential oil was G^+^ bacteria *B. subtilis* and yeast *C. albicans*. 

Table 5 also shows the effects of (*E*)-caryophyllene, in different concentrations, on the tested microbial strains. In general, using this method, treatment with the tested compound showed very high susceptibility to bacterial and yeast strains. The obtained MIC50 and MIC90 values were in the range of 0.22–0.75 µg/mL for MIC50 and 0.39–0.89 µg/mL for MIC90. The best antimicrobial activity was determined against G^+^ bacterial strains *E. faecalis* and *S. aureus*, while out of G^−^ bacteria, the most sensitive was biofilm-forming *P. fluorescens* with all three showing a MIC50 of 0.22 and a MIC90 of 0.39 µg/mL. Out of yeast strains, (E)-caryophyllene was most effective in inhibiting *C. albicans* and *C. tropicalis* with a MIC50 of 0.56 and a MIC90 of 0.67 µg/mL.

In order to compare the results of SSEO and its constituent (*E*)-caryophyllene to the effect of a standard antibiotic, we have used the same method to evaluate the activity of meropenem (Table 5). The best antimicrobial activity of this standard antibiotic was found against the G^−^ bacteria *Y. eneterocolitica* with a MIC50 of 0.73 and a MIC90 of 0.98 µg/mL. Out of G^+^ bacteria, the most sensitive to meropenem was *E. faecalis* with a MIC50 of 11.72 and a MIC90 of 13.74 µg/mL. Considering the yeast strain, *C. albicans* and *C. glabrata* were the most sensitive on the exposure to the meropenem, with MIC values of 2.93 µg/mL for MIC50 and 4.24 µg/mL for MIC90. 

Comparing the results obtained by using a minimal inhibitory concentration assay, the overall conclusion is that the tested microbial strains show the highest sensitivity to the treatment with (*E*)-caryophyllene. 

Because of the difficulties in evaluating mycelial growth using a spectrometer, a minimal inhibition concentration assay of microscopic fungi was evaluated with a different method. In this study, we measured inhibition zones at different concentrations of SSEO. Table 3 shows the effects of a concentrated SSEO, while Table 6 shows the effects of SSEO in different concentrations (500, 250, 125, and 62.5 µL/mL obtained by dilution in DMSO). The best antifungal activity of SSEO against *A. flavus* was observed at the applied concentration of 500 µL/mL (8.00 ± 3.00 mm), while the lowest activity was observed at a concentration of 125 µL/mL (1.67 ± 0.58 mm). Against *B. cinerea*, the best effect of SSEO was detected at the applied concentration of 250 µL/mL (9.67 ± 1.53 mm) and the lowest at the concentration of 125 µL/mL (5.67 ± 0.58 mm). The SSEO was most effective in inhibiting the growth of *P. citrinum* at a concentration of 250 µL/mL (7.33 ± 0.58), and least effective at a concentration of 62.5 µL/mL (5.33 ± 0.58). 

### 2.4. Antimicrobial Activity In Situ

In the next step of this research, we evaluated the antimicrobial potential of the SSEO vapor phase. The effects of different concentrations of SSO vapor phase were examined against, G^+^, G^−^ bacteria, and microscopic filamentous fungi growing on potatoes (Table 7). Generally, results indicate strong antibacterial effects of SSEO against all G^+^ and G^−^ bacteria. Out of the tested G^+^ bacteria strains, SSEO showed the best antimicrobial activity against *S. aureus* and *E. faecalis* at the applied concentration of 62.5 µL/L (73.71 ± 2.06% and 64.71 ± 0.78%, respectively). Interestingly, all G^+^ bacteria were the most resistant on the SSEO at the highest concentration applied (500 µL/L). Moreover, the probacterial effect of SSEO was detected in the case of *B. subtilis* at a concentration of 500 µL/L. Statistical analysis revealed significant differences between the applied concentrations of each bacterium. 

Considering G^−^ bacteria growing on the potato model, the highest percentage of effectiveness (74.79 ± 3.66%) was observed on the growth of *S. enterica* at a concentration of SSEO of 62.5 µL/L. Moreover, considerably strong antibacterial effects of SSEO in the lowest applied concentration of 62.5 µL/L were detected on the growth of *Y. enterocolitica* (67.61 ± 1.84%) and biofilm-forming *P. flourescens* (56.46 ± 1.33%), while the growth of *P. aeruginosa* was inhibited in the highest percentage by the SSEO in the concentration of 250 µL/L (67.74 ± 1.89%). Statistical analysis showed significant differences between concentrations applied to *P. flourescens* and *S. enetrica*. 

The best anti-yeast activity was found against *C. krusei* at the SSEO concentration of 250 µL/L (67.61 ± 0.86%) and *C. glabrata* at a concentration of 62.5 µL/L (64.01 ± 1.68%). *C. albicans* growing on potato modal was most effectively inhibited by the SSEO at a concentration of 62.5 µL/L (34.09 ± 1.25%), while at concentrations of 250 and 500 µL/L, probacterial activity was observed. Moreover, the growth of *C. tropicalis* was enhanced by the SSEO at the applied concentrations of 62.5 and 500 µL/L. Statistical analysis revealed significant differences between the applied concentrations of each yeast strain.

The results of the antifungal effects of the SSEO vapor phase on the potato model show strong activity against all microscopic fungi tested. The strongest activity was noted against *P. citrinum* (87.34 ± 2.22%), *B. cinerea* (73.00 ± 1.16%), and *A. flavus* (64.11 ± 1.15%) at the SSEO concentration of 62.5 µL/L. Statistical analysis revealed significant differences in each microscopic fungus between all concentrations applied.

### 2.5. Antibiofilm Activity

The effects of SSEO (experimental group) and the control (untreated) on the biofilm-producing *P. fluorescens* bacteria were evaluated using mass spectrometry over 14 days. In order to analyze the protein profile spectra of treated and untreated *P. fluorescens* biofilm-forming bacteria, we have used the MALDI-TOF MS Biotyper. Using this method changes in the molecular structure of the bacteria following growth can be observed. The experiment was performed on two different surfaces (stainless steel and plastic). Due to the identity of the development of the mass spectra of the control planktonic cells and the control mass spectra obtained from the surfaces on individual days, the control planktonic spectra were chosen as a representative of the development of the mass spectra of the control group on individual days. 

Owing to the addition of SSEO to the experimental group, changes in the mass spectra of both surfaces were recorded already on the 3rd day of the experiment (Figure 1A). In the case of the mass spectra obtained from the antibody surface of the experimental group, there was a decrease in the number of mass spectra peaks compared to the control planktonic spectra. On the contrary, in the experimental spectra obtained from the plastic surface, there was an increase in the number of detected peaks compared to the control planktonic spectrum. These results indicate no significant inhibition of the *P. fluorescens* biofilm-forming bacteria after the 3rd day of treatment with SSEO. However, results obtained on the 5th day of the experiment (Figure 1B) display an increase in the detected peaks in the mass spectra of the experimental groups compared to the control planktonic spectrum on the same day of the experiment. On the 7th day of the experiment (Figure 1C), a visible change in the experimental mass spectra compared to the control was observed. Based on these observations, we conclude that SSEO clearly initiated the inhibition of *P. fluorescens* biofilm. This trend continued until the end of the experiment (days from 9th to 14th), which can be seen in Figure 1D–F. Moreover, the most significant differences were observed on the last day of the experiment. 

From our observation of the development of the mass spectra, it follows that SSEO influenced the development of the biofilm of the experimental groups from the beginning of the experiment, but the proven inhibitory effect was detected from the 7th day of the experiment. Based on the analysis, we conclude that SSEO is suitable for inhibiting *P. fluorescens* biofilm.

A dendrogram constructed according to the mass spectra is presented in Figure 2. Based on the MSP distances, the constructed dendrogram reflected the similarities in the biofilm structure of the control and experimental groups. It can be observed from the dendrogram that the shortest MSP distances were reached by the early biofilm stages of the experimental groups, during days 3 and 5 (PFS 3; 5 and PFP 3; 5) together with the control groups (CPF 3-14). As the time of exposure of the experimental groups to SSEO increased, their MSP distance increased compared to the control groups. The most significant difference occurred on the 14th day of the experiment, when the MSP distances of the experimental groups were the largest, with the predominance of the experimental group on the stainless-steel surface (PFS 14). From this observation, we concluded that SSEO affects the homeostasis of the *P. fluorescens* biofilm, thereby helping to inhibit it. These findings are consistent with mass spectral analysis.

### 2.6. Insecticidal Activity of SSEO

Table 8 shows the insecticidal activity of SSEO on *O. lavaterae*. The best insecticidal activity was observed for the concentrated SSEO (100% concentration of SSEO). SSEO at the concentrations of 6.25 and 3.125% did not show any repellent effect against *O. lavaterae*, while SSEO at the concentration of 50% affected 50% of the *O. lavaterae* population and 25% of SSEO has an activity for 43.33% of insects.

## 3. Discussion

The chemical composition of volatiles presented in EO obtained from *S. sclarea* was determined by means of GC and GC-MS analysis. Our results indicate that this sample is characterized by a high amount of oxygenated monoterpenes linalool acetate (49.1%) and linalool (20.6%). Many previous reports regarding the chemical composition of *S. sclarea* showed very similar results to the one obtained in this study [2,8,28,29,30,31,32]. Differences between the volatiles in these publications are displayed in the contribution of the compounds in minor amounts. In some papers, the most abundant compounds besides linalool acetate and linalool are reported to be germacrene D [2,28], sclareol [2,30], geranyl acetate [28,30,32], α-terpineol [30,32], and geraniol [28,30]. However, in this term, our results defer, since the higher amounts identified were (*E*)-caryophyllene, *p*-cimene, and geranyl acetate. Reported variations in the chemical composition of SSEO can be attributed to different climatic and geographical conditions, soil characteristics, as well as the method of essential oil extraction. 

Concerning the results of inhibition of the DPPH radical and the ABTS radical cation, SSEO showed an insufficient effect regarding the standard compound. These data are in agreement with previously published results of this species with a high abundance of linalool and linalyl acetate [8], while other reports where the abundance of linalyl acetate was higher (over 60%) indicate better results of SSEO antioxidant activity [32]. This phenomenon can be explained by the synergistic and antagonistic actions of all compounds presented in the EO [8]. Moreover, our results indicate better neutralization of the ABTS radical cation compared to the DPPH radical, which is consistent with the previous findings that indicate that the examination of plant foods containing hydrophilic, lipophilic, and high-pigmented antioxidant compounds shows the superiority of the ABTS assay compared to the DPPH [33]. 

In order to determine the antimicrobial effectiveness of SSEO and its constituent (*E*)-caryophyllene, we used different methods. The preliminary step in the evaluation of antimicrobial activity was the disc diffusion method, which in general revealed better activity of SSEO compared to the (*E*)-caryophyllene standard compound. Nevertheless, the biggest zones of inhibition were noted for the standard antibiotics meropenem (for bacteria) and fluconazole (for microscopic filamentous fungi). On the other hand, the results of the broth microdilution method showed the best activity of (*E*)-caryophyllene compared to SSEO and the standard antibiotic meropenem. Previous reports obtained by disc diffusion assay showed that *S. sclarea* EO can effectively inhibit the growth of G^+^ and G^−^ bacterial strains [32,34]. Ovidi et al. reported that using this method, SSEO was active against *A. bohemicus*, *K. marina*, and *B. cereus* [32]. Results obtained in our study using the disc diffusion method show that SSEO can strongly inhibit the growth of *B. subtilis*, *C. albicans*, and *A. flavus*. However, results published until now obtained by the broth microdilution method differ. Even though our results indicate that SSEO displays the strongest inhibition of G^+^
*S. aureus* growth, a general conclusion can be made that G^−^ bacterial strains tested in our study were more sensitive to the treatment with this EO. Aćimović et al. also reported that G^+^ bacterial strains were more resistant to exposure to *S. sclarea* EO compared to the G^−^ bacteria. In the same study, *E. coli* and *S. enteritidis* were the most sensitive to the treatment with SSEO [8]. However, the report made by Cui et al. showed no differences between the inhibition of G^+^ and G^−^ bacterial strains after treatment with SSEO, and the overall conclusion of this study is that this EO is an effective bacterial inhibitor and bactericide with a broad antibacterial spectrum [2]. Another study showed that this EO displayed no activity against *P. fluorescens*, which differs from the results obtained in this study that indicate moderate inhibition of this bacteria by the results of the disc diffusion method and MIC values of 2.93 for MIC50 and 3.17 for MIC90 [32]. The same study, however, reported the highest susceptibility of *E. coli* to the treatment with SSEO, which is in agreement with the findings obtained by Kuzma et al. [28]. Kuzma et al. also reported that *S. aureus* and *S. epidermidis* were sensitive to the treatment with SSEO [28]. Previously reported tests of the antifungal activity of this EO showed a dose-dependent inhibition of mycelial growth of *S. sclerotiorum*, *S. cepivorum*, and *F. oxysporum* f. sp. *dianthi* [31]. Considering the yeast strains, previous reports indicate that SSEO showed antifungal activity against clinical isolates of the genus Candida, with *C. albicans* being the most sensitive strain and *C. glabrata* the most resilient one [35]. The results obtained in this study differ in that both of these strains were the most resilient to the effects of SSEO. The differences between published results can be attributed to the differences among microbial species and strains of the same origin examined, the variation of material for extraction of *S. sclarea* EO, which implies the diversity of chemical constituents responsible for the detected activity, as well as their synergistic effect, etc. [2,8]. 

Previous reports regarding the antimicrobial activity of (*E*)-caryophyllene suggest strong antibacterial effects and also significant activity towards fungi strains [36,37]. These publications also agree on the higher sensitivity of G^+^ compared to G^−^ bacterial strains in the treatment with this compound. Schmidt et al. also reported that (*E*)-caryophyllene showed no inhibition of *P. aeruginosa*, which disagrees with our results that indicate strong inhibition of this bacterial strain using the broth microdilution method and moderate activity using the disc diffusion method [37]. Literature data also shows that (*E*)-caryophyllene can be used as an antimicrobial agent against periodontopathogens [38,39]. 

Meropenem is a well-known, broad-spectrum antibiotic that has been used in clinical practice for nearly 25 years. Its activity has been observed on many Gram-positive and Gram-negative bacterial strains. Previously published studies showed that this antibiotic has very high activity (99.1%) and the potential as an alternate therapy in the treatment of prevalent pathogens such as MDR G^−^ bacilli and endemic *Salmonella* spp. [40]. Likewise, Joly-Guillou [41] and associates showed that meropenem displays a cumulative susceptibility rate against *E. coli* of 100% and against other *Enterobacteriaceae* of 99%. Kobayashi et al. investigated the antimicrobial activity of meropenem against a total of 187 bacteria isolates, including 43 *E. coli*, 23 *K. pneumoniae*, 9 *E. cloacae*, 22 *P. aeruginosa*, 16 methicillin-susceptible *S. aureus* (MSSA), 24 methicillin-resistant *S. aureus* (MRSA), and 50 *S. epidermidi*, from patient blood in Keio, and their results showed that meropenem has potent and stable antibacterial effects against G^−^ bacteria strains [42]. The results obtained in this study agree with the previous findings that this antibiotic has strong effects on the tested bacterial strains with inhibition zones that were found to be in the range of 27 ± 2.0 mm to 35 ± 1.0 mm for the disc diffusion assay. However, results obtained by using the broth microdilution method show that meropenem displays moderate effects. The best activity of this antibiotic was observed for G^−^ bacteria *Y. enterocolitica* and yeast strains *C. albicans*, *C. glabrata*, and *C. krusei*. High resistance to the treatment with meropenem was observed for the G^−^ bacteria *P. aeruginosa* and the yeast strain *C. tropicalis*. 

EOs have already been identified as natural food additives useful for applications in food preservation. Considering that, their application in fighting against foodborne microbes and spoilage microorganisms represents a novel approach in antimicrobial research [43,44,45]. Mainly, studies made up until now have focused on the antibacterial mechanisms, including direct contact type and fresh-keeping effects of EOs. In those studies, the EOs were generally spread onto the surface of the food or coated into the packaging material, which implies direct contact and therefore affects their organoleptic properties [46,47,48,49,50]. However, literature data suggest that, in order to exert antimicrobial effects, EOs do not require direct contact since they occur as well in the vapor phase, which is highly bioactive [51]. Moreover, in the liquid phase, higher concentrations of EOs are required to achieve effective antimicrobial activity compared to the vapor phase [45]. Some previous studies have also shown that some EOs are better antifungals in the vapor phase compared to their liquid phase [45,51]. This phenomenon can be explained by the tendency of fungal strains to grow more than bacteria on the agar surface, exposing them more to the vapor. Moreover, there is a factor of lipophilicity in the molecules responsible for the expressed beneficial effects, which possess the ability to associate and make micelles and suppress their attachment to the organism [45]. On the other hand, the vapor phase permits free attachment [45]. Considering the above mentioned, in this study we have evaluated the activity of the SSEO vapor phase against G^+^ and G^−^ bacteria and microscopic filamentous fungi growing on potatoes. Overall, the obtained results indicate the best antimicrobial activity of SSEO in a potato model with microorganisms at the lowest concentration applied. Interestingly, in some cases, our results indicate that higher concentrations of SSEO can induce microbial growth. Generally, the strong activity of SSEO has been observed on all tested microorganisms, but the strongest inhibition effects have been observed for *S. aureus*, *E. faecalis*, *S. enterica*, and *Y. enterolitica* from the bacterial strains and *C. glabrata* from the yeasts. As for the fungi strains, SSEO effectively inhibited all the tested strains. In conclusion, the essential oil analyzed in this study showed significantly stronger antimicrobial activity in the vapor phase, which is probably due to the significant proportion of volatile compounds that act in synergy. These results are in agreement with the initial hypothesis that the vapor phase of essential oils shows better effects compared to direct contact. 

Biofilm resistance to the treatment with antibiotics has attracted the great attention of the scientific community. The biofilms are regarded as microorganism communities that adhere to a biotic or abiotic surface, which distinguishes them from planktonic cells [52,53]. Moreover, the formation of biofilms is considered the usual mode of bacterial growth, while a planktonic, single-celled state corresponds to a transitional phase. Literature data show that biofilm formation is influenced by characteristics of the contact surface, or bacterial cell, growth medium, and environmental factors [54]. Investigations made up until now imply that over 65% of all infectious diseases and chronic infections involve biofilms. Moreover, concerning is the fact that biofilm formation increases drug tolerance 10 to 1000-fold in comparison to planktonic bacteria [55,56]. For this reason, it is crucial to investigate new agents that can efficiently inhibit or eradicate bacterial biofilm formation. One of the possible alternatives imposed is the use of EOs because of their diverse modes of action in bacterial growth inhibition, which prevents microorganisms’ resistance [57,58]. Some previous studies indicate that these volatile plant extracts inhibit the effects of biofilm formation at concentrations below their minimal inhibitory concentrations [59]. So, with the aim to further investigate the displayed strong antimicrobial effects of SSEO, we have evaluated its inhibitory effect on the protein profile of *P. flourescens* biofilm formed on stainless-steel and plastic surfaces. Abnormalities in protein structure correlated with biofilm formation and degradation after exposure to EOs can be monitored by using MALDI-TOF mass spectra, a method used in this study. Results obtained clearly indicate that treatment with SSEO on the 7th day of the experiment initiated the inhibition of the biofilm-forming *P. flourescens* on both surfaces. This trend continued to the last day of the experiment (14th day) when the most significant inhibition was observed. The presented results suggest that treatment with SSEO affected the homeostasis of *P. flourescens* bacterial biofilms that were formed on stainless-steel and plastic surfaces. Previous results showed that SSEO can be considered for the treatment of chronic infections caused by *P. aeruginosa* biofilms [60]. 

Botanical insecticides represent naturally occurring plant-derived insecticides. Resistance of insects to conventional chemicals has increased the interest in these types of insecticides in the last 15 years [61]. This led to an increase in investigations of essential oils and plant extracts as insecticidals against different stored-product pests [62,63,64,65]. Even though interest in botanical insecticides has increased, only a few of the products for pest control that are in use are directly obtained from plants [66]. According to the literature data, botanical insecticides represent only 1% of the world insecticide market [67]. In this study, the insecticidal activity of SSEO against *O. lavaterae* was investigated. The strongest activity was observed at the highest concentration applied (100% concentration of SSEO). In a concentration of 50%, SSEO affected 50% of the *O. lavaterae* population, while the concentration of 25% of SSEO affected 43.33% of insects. To our knowledge, this is the first report of *S. sclarea* essential oil insecticidal activity.

## 4. Materials and Methods

### 4.1. Essential Oils and Standard

Clary sage (*Salvia sclarea*) essential oil was purchased from the Slovak company Hanus s.r.o. (Nitra, Slovakia). The essential oil was obtained by steam distillation of young branches and leaves of *Salvia sclarea* by the producer. The standard compound, (*E*)-caryophyllene, ≥98.5% was purchased from Sigma-Aldrich (Taufkirchen, Germany).

### 4.2. Gas Chromatography–Mass Spectrometry and Gas Chromatography Analyses 

Using an Agilent Technologies (Palo Alto, Santa Clara, CA, USA) 6890 N gas chromatograph equipped with a 5975 B quadrupole mass spectrometer (Agilent Technologies, Santa Clara, CA, USA), we have performed a chemical constituent analysis of *S. sclarea* essential oil. The separation of volatiles was carried out on an HP-5MS capillary column (30 m × 0.25 mm × 0.25 µm). The Agilent Technologies gas chromatograph was operated by an interfaced HP-Enhanced ChemStation software D.03.00.611 (Agilent Technologies). The diluted essential oil sample was (10% hexane solution) injected in a volume of 1 µL, with helium 5.0 used as a carrier gas (flow rate of 1 mL/min). The temperature of the MS source and MS quadruple were set at 230 °C and 150 °C, respectively, while the temperature of the split/splitless injector was set at 280 °C. The mass scan was in the range of 35–550 amu at 70 eV, and the solvent delay time was 3.00 min. The chromatographic conditions of GC and GC-MS analysis were as follows: temperature program of 60 °C to 150 °C (rate of increase 3 °C/min), and 150 °C to 280 °C (rate of increase 5 °C/min), held 4 min at 280 °C, and the total run time was 60 min. The split ratio was 40.8:1. 

Identification of volatile constituents was performed by comparison of their retention indices (RI) as well as the reference spectra reported in the literature and the ones stored in the MS library (Wiley7Nist) [68,69]. Using GC-FID with the same HP-5MS capillary column, semi-quantification of the components was performed, taking into consideration amounts higher than 0.1%.

### 4.3. Antioxidant Activity

The radical scavenging of 2,2-diphenyl-1-picrylhydrazyl (DPPH, Sigma-Aldrich, Taufkirchen, Germany) and 2,2′-azino-bis(3-ethylbenzothiazoline-6-sulfonic acid) (ABTS Sigma-Aldrich, Taufkirchen, Germany) were used to measure the antioxidant activity of SSEO. DPPH was dissolved in methanol at a concentration of 0.025 g/L. Prior to analysis, the absorbance of DPPH was adjusted to 0.8 at wavelength 515 nm (by Glomax spectrophotometer, Promega Inc., Madison, WI, USA). The ABTS radical cation was generated according to the previously described method [70]. The prepared radical cation was diluted before the analysis up to an absorbance value of 0.7 at 744 nm. The 190 μL of the DDPH or ABTS solution was mixed with 10 μL of SSEO (in a 96-well microtiter plate) for 30 min with continuous shaking at 1000 rpm at room temperature in the dark. For the ABTS assay, a decrease in absorbance at 744 nm was registered using a microplate reader. A decrease in absorbance at 515 nm was registered for the DPPH assay. The percentage of DPPH or ABTS inhibition was calculated according to the formula (A0 − AA)/A0 × 100, where A0 was the absorbance of DPPH or ABTS with methanol and AA was the absorbance of the sample. The standard reference Trolox (Sigma-Aldrich, Schnelldorf, Germany) was used to calculate the total antioxidant capacity. Trolox was dissolved in methanol (Uvasol^®^ for spectroscopy, Merck, Darmstadt, Germany) to a concentration range of 0–100 µg/mL. Total antioxidant activity was expressed according to the Trolox calibration curve (TEAC). 

### 4.4. Microorganisms

Three Gram-positive bacteria (*Bacillus subtilis* CCM 1999, *Enterococcus faecalis* CCM 4224, and *Staphylococcus aureus* subsp. *aureus* CCM 8223), four Gram-negative bacteria (*Pseudomonas aeruginosa* CCM 3955, *Salmonella enterica* subsp. *enterica* CCM 4420, *Yersinia enterocolitica* CCM 7204, and biofilm-forming *P. flourescens*), and four yeasts (*Candida albicans* CCM 8261, *Candida glabrata* CCM 8270, *Candida krusei* CCM 8271, and *Candida tropicalis* CCM 8223), which were obtained from the Czech Collection of Microorganisms (Brno, Czech Republic) and three microscopic fungi (*Aspergillus flavus*, *Botrytis cinerea*, *Penicillium citrinum*) were used for antimicrobial activity. The biofilm-forming bacterial strain *Pseudomonas fluorescens* was used for analyses of antibiofilm activity. *P. fluorescens* was isolated from freshwater fish *Abramis bram*, the skin part. The fish was obtained from the retail market in Latvia. Microscopic fungi *A. flavus*, *B. cinerea*, and *P. citrinum* were isolated from the grape. Biofilm-forming bacteria and fungi were identified by 16S rRNA sequencing and the MALDI-TOF MS Biotyper. 

### 4.5. Disc Diffusion Method

The disc diffusion method was used for the preliminary evaluation of the antimicrobial activity of SSEO and its constituent (*E*)-caryophyllene. Bacterial inoculum was cultivated for 24 h on Tryptone soya agar (TSA, Oxoid, Basingstoke, UK), while yeast inoculum was cultivated on Sabouraud dextrose agar (SDA, Oxoid, Basingstoke, UK). Bacteria and yeast inoculums were kept at 37 °C and 25 °C, respectively. The microbial culture was adjusted using distilled water to an optical density of 0.5 McFarland standard (1.5 × 10^8^ CFU/mL). One hundred microliters of bacterial and microscopic filamentous fungi cultures was spread on Mueller–Hinton agar (MHA, Oxoid, Basingstoke, UK) (for bacteria) or Sabouraud dextrose agar (SDA, Oxoid, Basingstoke, UK) (for fungi) [71]. Next, sterile 6 mm discs saturated with 10 μL of SSEO and 10 mg/mL of (*E*)-caryophyllene were placed on microbial suspension, and plates were incubated at 37 °C for bacteria and 25 °C for microscopic fungi in a duration of 24 h. Inhibition zones were measured on three sides from the edge of the filter, and results were interpreted in the following manner: very strong antimicrobial activity for inhibition zones larger than 10 mm; moderate activity 10–5 mm; and inhibition zone 5–1 mm was determined as weak activity. The antibiotics used as controls were meropenem (Oxoid, Basingstoke, UK) for bacteria and fluconazole (Oxoid, Basingstoke, UK) for microscopic filamentous fungi. The method for evaluating the inhibition zones of biofilm-forming bacteria was the same. All measurements were performed in triplicate. 

### 4.6. Broth Microdilution Method

Bacterial inoculum was cultivated for 24 h in Mueller–Hinton broth (MHB, Oxoid, Basingstoke, UK) at 37 °C, while yeast inoculum was cultivated on Sabouraud dextrose broth (SDA, Oxoid, Basingstoke, UK) at 25 °C. A volume of 50 μL of inoculum with an optical density of 0.5 McFarland standard was added to a 96-well microplate. SSEO at a volume of 100 μL was added to the microbial suspension at final concentrations ranging from 500 μL/mL to 0.244 μL/mL. Samples were mixed and incubated for 24 h at 25 °C (yeast cultures) and 37 °C (bacterial cultures) [71]. 

The (*E*)-caryophyllene and standard antibiotic meropenem (10 mg) were dissolved in 0.1% DMSO (Sigma-Aldrich, Taufkirchen, Germany) and diluted so the final concentrations ranged from 500 μg/mL to 0.244 μg/mL.

Seven-day-old cultures grown on agar plates (CYA) were used for the preparation of the mold conidia suspensions. Conidia suspensions were prepared in a sterile saline solution. The turbidity of the suspension was adjusted with a spectrophotometer  (densilameter II, Erba-Lachema, Brno, Czech Republic) at 530 nm to obtain a final concentration that matches that of a 0.5 McFarland standard. Briefly, 100 µL of spore suspension was spread thoroughly all over the surface of Sabouraud dextrose agar (SDA; Hi-Media Laboratory, Maharashtra, India) plates. The plates were dried in an air-dry stiller at 60 °C until evaporation of residual water. Sterile paper discs (6 mm in diameter; Oxoid, Cambridge, UK) were impregnated with 20 µL of SSEO at a desired concentration (500, 250, 125, and 62.5 µh/mL/disc) and deposited on the agar surface. The test for antifungal properties of SSEO was performed in triplicate, for each microorganism and each concentration. The Petri dishes were incubated at 25 ± 1 °C, for 24 h in a thermostat (Friocell, MMM Medcenter Einrichtungen GmbH, Planegg, Germany). After 24 h of the incubation period, the antifungal agent diffused into the agar and inhibited the germination and growth of the tested microorganism. The diameters of inhibition growth zones were measured as semidiameters (in millimeters). Pure DMSO and fluconazole were used as controls for each tested fungus. 

### 4.7. Analysis of Biofilm Degradation

Degradation of the protein spectra during biofilm development was evaluated for biofilm-forming *P. fluorescens.* Various phases of protein spectra were recorded using the MALDI-TOF MS Biotyper. The experimental and control samples were prepared in polypropylene tubes. Briefly, 20 mL of MHB was added to 50 mL tubes along with microscopic slides made of plastic and stainless steel and 200 μL of biofilm-forming bacteria. SSEO was added to experimental groups at a final concentration of 0.1%, and control samples were left untreated. The samples were incubated at 37 °C on a shaker at 170 rpm and analyzed on days 3, 5, 7, 9, 12, and 14. The biofilm samples were taken from a plastic and stainless-steel surface with a sterile cotton swab and placed on a MALDI-TOF target plate. The planktonic cells were taken from 300 µL of culture medium. The bacterial suspension was centrifuged for 1 min at 12,000 rpm, and the supernatant was discarded. The pellet was three times washed in 30 μL of ultrapure water and centrifuged. After the washing, planktonic cells were resuspended, and 1 μL was applied to a target plate. Next, 1 μL of α-Cyano-4-hydroxycinnamic acid matrix (10 mg/mL) was applied to the dried target plate with samples. MALDI-TOF MicroFlex (Bruker Daltonics) was used for the analysis of biofilm protein structure. The spectra were recorded in linear and positive modes with a mass-to-charge ratio of 200–2000. The protein spectra were obtained by automatic analysis, and the similarities of the spectra in one sample were used to generate the standard global spectrum (MSP). Nineteen MSP were generated from the spectra by MALDI Biotyper 3.0 and grouped into dendrograms using Euclidean distance [72].

### 4.8. Insecticidal Activity

The insecticidal activity of SSEO was evaluated on a model organism, *Oxycarenus lavaterae*. Thirty *O. lavaterae* individuals were placed in the PD in each group. A circle of sterile filter paper was glued to the lid. Concentrations (100, 50, 25, 12.5, 6.25, and 3.125%) were prepared by diluting SSEO with 0.1% polysorbate. One hundred microliters of the appropriate concentration of SSEO was applied to the sterile filter paper. The dishes were sealed around the perimeter with parafilm and left at room temperature for 24 h. In the control group, 100 µL of 0.1% polysorbate was used. After 24 h, the number of living and dead individuals was evaluated. The experiment was performed in triplicate.

### 4.9. Statistical Data Evaluation

All measurements and analyses were carried out in triplicate. The mean and standard deviation (SD) were calculated using Microsoft Excel software 2020. One-way analysis of variance (ANOVA) was performed using Prism 8.0.1 (GraphPad Software, San Diego, CA, USA) at the 5% level of significance. From the measured absorbances obtained before and after this experiment, we converted the differences in absorbance between measurements to a set of binary values. These values were assigned exact concentrations. For this experiment, we created the following formula: if absorbance values were as low as 0.01, then the numbers for the binary system were 1 (inhibitory effect); if absorbance values were as high as 0.01, then numbers for the binary system were 0 (no effect or stimulant effect). For this statistical evaluation, Probit analysis in Statgraphics software 19 was used. The method was modified by the authors, Vatľak et al. [73].

## 5. Conclusions

The main compounds of *S. sclarea* essential oil examined in this study were linalool acetate (49.1%) and linalool (20.6%), followed by high amounts of (*E*)-caryophyllene (5.1%), *p*-cymene (4.9%), α-terpineol (4.9%), and geranyl acetate (4.4%). The antioxidant activity assays showed insufficient antioxidant effects of SSEO compared to the Trolox standard, with a neutralization potency of 11.76 ± 1.34% for the DPPH radical and 29.70 ± 1.45% for the ABTS radical cation. Regarding antimicrobial activity, the EOs of *S. sclarea*, (*E*)-caryophyllene, and meropenem had generally moderate effects. Using the disc diffusion method, the results indicated better effects of the standard antibiotics compared to the treatment with SSEO and its component (*E*)-caryophyllene. Fluconazole and meropenem showed inhibition diameters in the range from 27 ± 1.5 to 35 ± 0.5 mm for all tested microorganisms, while SSEO was the most effective against the G^+^ bacterial strain *B. subtilis* with an inhibition zone of 12 ± 1.00 mm, and for (*E*)-caryophyllene, the best antimicrobial activity was found against the yeast *C. krusei* with an inhibition zone of 7.67 ± 0.58 mm. On the other hand, results obtained by employing a minimal inhibitory concentration assay revealed that (*E*)-caryophyllene has the potency to inhibit the growth of tested microbial strains to a greater extent compared to the standard antibiotic meropenem and tested EO. The minimal inhibition values for (*E*)-caryophyllene were found to be in the range of 0.22–0.75 µg/mL for MIC50 and 0.39–0.89 µg/mL for MIC90. SSEO showed the best antimicrobial effects against G^+^ against *S. aureus* (MIC50 of 1.48 and MIC90 of 1.59 µL/mL), while meropenem was the most effective against G^−^ bacteria *Y. eneterocolitica* with a MIC50 of 0.73 and a MIC90 of 0.98 µg/mL. The antimicrobial testing of SSEO in the vapor phase demonstrated significantly high inhibition of microbial strains growing on the potatoes model. In general, this study demonstrated higher sensitivity of microorganisms at lower concentrations of oil applied. SSEO has also shown strong activity in inhibiting the biofilm formation of *P. fluorescens* growing on plastic and stainless-steel surfaces, affecting the homeostasis of the bacteria, as observed by the MALDI-TOF MS Biotyper. Evaluation of insecticidal activity revealed that SSEO at a concentration of 100% was able to affect more than 66% of *O. lavaterae*. 

On the basis of our findings, we hypothesize that SSEO and (*E*)-caryophyllene could find application as potential antimicrobial agents. Moreover, SSEO can be effective in the fight against biofilms in various industries, and it can be a promising agent for prolonging the storage and shelf-life extension of potatoes. 

## Figures and Tables

**Figure 1 ijms-24-05179-f001:**
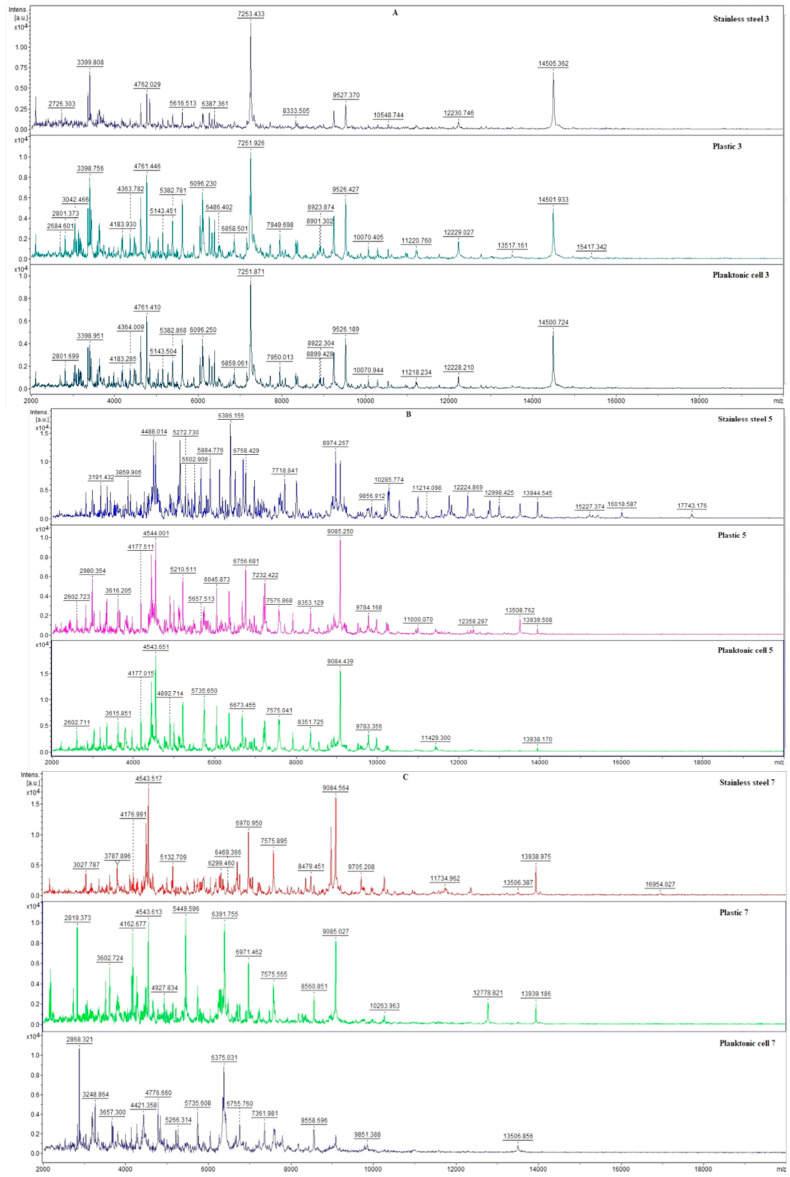
MALDI-TOF mass spectra of *P. fluorescens* biofilm development after SSEO exposition: (**A**)—3rd day; (**B**)—5th day; (**C**)—7th day; (**D**)—9th day; (**E**)—12th day; (**F**)—14th day.

**Figure 2 ijms-24-05179-f002:**
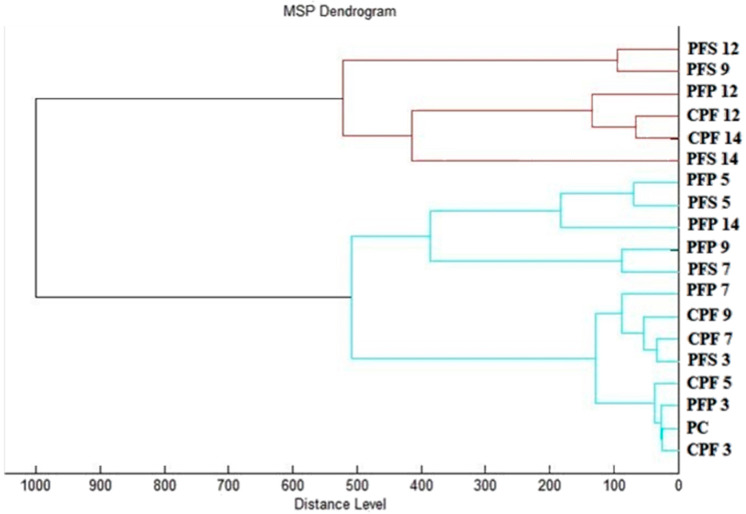
Dendrogram of *P. fluorescens* biofilm progress after SSEO exposition. PF—*P. fluorescens*; C—control; S—stainless-steel; P—plastic; P—planktonic cells.

**Table 1 ijms-24-05179-t001:** Chemical composition of SSEO.

No	RI(calc) ^a^	RI(lit)	Compound ^b^	%
1	858	859	cis-3-hexenol	tr ^c^
2	938	939	α-pinene	2.4
3	948	954	camphene	tr
4	977	975	sabinene	tr
5	980	979	β-pinene	0.2
6	992	990	β-myrcene	0.6
7	1004	1002	α-phellandrene	tr
8	1009	1011	δ-3-carene	tr
9	1016	1010	α-terpinene	tr
10	1023	1024	*p*-cymene	4.9
11	1028	1029	α-limonene	2.2
12	1038	1037	(*Z*)-β-ocimene	0.3
13	1047	1050	(*E*)-β-ocimene	0.5
14	1074	1072	cis-linalool oxide	tr
15	1088	1088	α-terpinolene	tr
16	1089	1086	trans-linalool oxide	tr
17	1098	1096	linalool	20.6
18	1189	1188	α-terpineol	4.9
19	1227	1229	nerol	1.1
20	1238	1238	neral	tr
21	1255	1257	linalool acetate	49.1
22	1286	1285	bornyl acetate	0.6
26	1299	1298	geranyl formate	tr
27	1364	1361	neryl acetate	1.7
28	1379	1375	α-copaene	0.2
29	1380	1381	geranyl acetate	4.4
30	1385	1388	β-bourbonene	tr
31	1388	1390	β-elemene	0.2
32	1408	1409	α-gurjunene	tr
33	1422	1419	(*E*)-caryophyllene	5.1
34	1456	1454	α-humulene	tr
35	1483	1481	germacrene D	0.2
36	1498	1496	ledene	tr
37	1502	1500	bicyclogermacrene	0.2
38	1525	1523	δ-cadinene	tr
39	1583	1583	caryophyllene oxide	0.3
	total			99.6

^a^ Values of calculated retention indices obtained experimentally on HP-5MS column; ^b^ identified compounds; ^c^ tr—compounds identified in amounts less than 0.1%.

**Table 2 ijms-24-05179-t002:** Percentage composition of each class of identified compounds.

Class of Compounds	%
Monoterpenes	93.5
Monoterpene hydrocarbons	11.1
Oxygenated monoterpenes	82.4
Monoterpene epoxide	tr ^a^
Monoterpene alcohols	26.6
Monoterpene aldehydes	tr
Monoterpene esters	55.8
Sesquiterpenes	6.1
Sesquiterpene hydrocarbons	5.9
Oxygenated sesquiterpenes	0.2
Sesquiterpene alcohols	tr
Sesquiterpene epoxides	0.3
Non-terpenic	tr
Alcohols	tr
Total	99.6

^a^ tr—compounds identified in amounts less than 0.1%.

**Table 3 ijms-24-05179-t003:** Antimicrobial activity of SSEO in mm.

Microorganism	Inhibition Zone	Activity of EO	Control
Gram-positive bacteria			
*Bacillus subtilis*	12.00 ± 1.00	***	33 ± 1.0
*Enterococcus faecalis*	4.67 ± 0.58	*	29 ± 0.5
*Staphylococcus aureus*	6.67 ± 0.58	**	32 ± 1.0
Gram-negative bacteria			
*Pseudomonas aeruginosa*	8.00 ± 1.00	**	25 ± 1.0
*Salmonella enterica*	3.67 ± 0.58	*	27 ± 2.0
*Yersinia enterocolitica*	3.67 ± 0.58	*	27 ± 1.5
*Pseudomonas fluorescens* biofilm	7.67 ± 0.58	**	28 ± 1.0
Yeasts			
*Candida albicans*	11.33 ± 0.58	***	28 ± 2.0
*Candida glabrata*	8.33 ± 0.58	**	33 ± 1.5
*Candida krusei*	7.67 ± 0.58	**	33 ± 3.0
*Candida tropicalis*	7.67 ± 0.58	**	33 ± 1.0
Fungi			
*Aspergillus flavus*	10.33 ± 0.58	**	32 ± 0.58
*Botrytis cinerae*	9.67 ± 0.58	**	33 ± 1.0
*Penicillium citrinum*	8.67 ± 0.58	**	31 ± 0.58

* Weak activity (zone 1–5 mm); ** moderate activity (zone 5–10 mm); *** strong activity (over 10 mm); antibiotics used as a control: cefoxitin for G^−^ bacteria, gentamicin for G^+^ bacteria, fluconazole for microscopic filamentous fungi.

**Table 4 ijms-24-05179-t004:** Antimicrobial activity of (*E*)-caryophyllene in mm.

Microorganism	Inhibition Zone	Activity	Control
Gram-positive bacteria			
*Bacillus subtilis*	3.67 ± 0.58	*	35 ± 0.5
*Enterococcus faecalis*	5.33 ± 0.58	**	34 ± 0.5
*Staphylococcus aureus*	5.67 ± 0.58	**	33 ± 1.0
Gram-negative bacteria			
*Pseudomonas aeruginosa*	5.33 ± 1.00	**	35 ± 1.0
*Salmonella enterica*	5.67 ± 0.58	**	34 ± 2.0
*Yersinia enterocolitica*	6.67 ± 0.58	**	34 ± 1.5
*Pseudomonas fluorescens* biofilm	5.67 ± 0.58	**	33 ± 1.0
Yeasts			
*Candida albicans*	5.33 ± 0.58	**	34 ± 2.0
*Candida glabrata*	5.33 ± 0.58	**	31 ± 1.5
*Candida krusei*	7.67 ± 0.58	**	33 ± 2.0
*Candida tropicalis*	5.67 ± 0.58	**	33 ± 1.0
Fungi			
*Aspergillus flavus*	5.33 ± 0.58	**	28 ± 1.0
*Botrytis cinerae*	5.67 ± 0.58	**	31 ± 1.0
*Penicillium citrinum*	5.67 ± 0.58	**	28 ± 1.5

* Weak activity (zone 1–5 mm); ** Moderate activity (zone 5–10 mm); antibiotics used as a control: meropenem for bacteria, fluconazole for microscopic filamentous fungi.

**Table 5 ijms-24-05179-t005:** Minimal inhibition concentration of SSEO, (*E*)-caryophyllene, and meropenem.

Microorganism	SSEO		(*E*)-Caryophyllene		Meropenem	
	MIC50	MIC90	MIC50	MIC90	MIC50	MIC90
	µL/mL		µg/mL		µg/mL	
Gram-positive bacteria						
*Bacillus subtilis*	187.31	199.21	0.37	0.44	23.44	4.24
*Enterococcus faecalis*	46.89	50.07	0.22	0.39	11.72	13.74
*Staphylococcus aureus*	1.49	1.59	0.22	0.39	93.80	98.56
Gram-negative bacteria						
*Pseudomonas aeruginosa*	46.89	50.07	0.37	0.44	187.31	199.87
*Salmonella enterica*	6.56	25.46	0.37	0.44	93.80	98.56
*Yersinia enterocolitica*	31.02	53.60	0.37	0.44	0.73	0.98
*Pseudomonas fluorescens* biofilm	2.93	3.17	0.22	0.39	93.80	98.56
Yeasts						
*Candida albicans*	131.99	153.98	0.56	0.67	2.93	4.24
*Candida glabrata*	31.02	53.60	0.75	0.89	2.93	4.24
*Candida krusei*	11.72	12.58	0.75	0.89	5.86	7.89
*Candida tropicalis*	2.93	3.17	0.56	0.67	298.92	324.56

**Table 6 ijms-24-05179-t006:** Minimal inhibition concentration of microscopic fungi in mm.

Fungi	Concentration of SSEO	Inhibition Zone in mm
*Aspergillus flavus*	500 µL/mL	8.00 ± 3.00 ^a^
	250 µL/mL	3.33 ± 0.58 ^b,a^
	125 µL/mL	1.67 ± 0.58 ^c,a^
	62.5 µL/mL	2.33 ± 0.58 ^d,a^
*Botrytis cinerea*	500 µL/mL	8.67 ± 0.58 ^a^
	250 µL/mL	9.67 ± 1.53 ^b^
	125 µL/mL	5.67 ± 0.58 ^c,a,b^
	62.5 µL/mL	6.67 ± 0.58 ^d,b^
*Penicillium citrinum*	500 µL/mL	6.00 ± 1.00 ^a^
	250 µL/mL	7.33 ± 0.58 ^b,a^
	125 µL/mL	6.00 ± 1.00 ^c,b^
	62.5 µL/mL	5.33 ± 0.58 ^d,b^

One-Way ANOVA, individual letters ^(a–d)^ in the upper case indicate the statistical differences between the concentrations.

**Table 7 ijms-24-05179-t007:** In situ analysis of the antimicrobial activity of the vapor phase of SSEO in potato.

Bacteria		Bacterial Growth Inhibition (%)			
		The Concentration of SSEO			
		62.5 μL/L	125 μL/L	250 μL/L	500 μL/L
Gram-positive	*B. subtilis*	24.86 ± 2.56 ^b^	46.85 ± 1.52 ^d^	32.34 ± 2.08 ^c^	−11.59 ± 0.90 ^a^
	*E. faecalis*	64.71 ± 0.78 ^d^	42.22 ± 1.36 ^c^	24.85 ± 2.66 ^b^	15.27 ± 1.62 ^a^
	*S. aureus*	73.71 ± 2.06 ^d^	53.98 ± 2.66 ^c^	24.07 ± 2.53 ^b^	12.75 ± 0.96 ^a^
Gram-negative	*P. flourescens biofilm*	56.46 ± 1.33 ^d^	46.85 ± 1.52 ^c^	24.92 ± 1.68 ^b^	15.04 ± 0.57 ^a^
	*P. aeroginosa*	25.37 ± 1.58 ^b^	8.48 ± 1.51 ^a^	67.74 ± 1.89 ^c^	7.77 ± 1.94 ^a^
	*S. enterica*	74.79 ± 3.66 ^d^	31.87 ± 1.48 ^b^	54.66 ± 1.82 ^c^	7.56 ± 1.09 ^a^
	*Y. enterocolitica*	67.61 ± 1.84 ^c^	43.78 ± 1.89 ^b^	12.26 ± 1.51 ^a^	64.87 ± 2.60 ^c^
Yeasts	*C. albicans*	34.09 ± 1.25 ^d^	8.71 ± 0.84 ^c^	−7.96 ± 1.35 ^b^	−12.37 ± 1.40 ^a^
	*C. glabrata*	64.01 ± 1.68 ^d^	44.59 ± 1.20 ^c^	33.66 ± 1.98 ^b^	14.92 ± 1.10 ^a^
	*C. krusei*	43.71 ± 0.95 ^c^	67.61 ± 0.86 ^d^	16.94 ± 1.54 ^b^	7.89 ± 1.45 ^a^
	*C. tropicalis*	−6.25 ± 0.65 ^b^	14.20 ± 1.56 ^c^	44.66 ± 1.20 ^d^	−15.26 ± 2.62 ^a^
Microscopic fungi	*A. flavus*	64.11 ± 1.15 ^d^	35.88 ± 1.06 ^c^	7.11 ± 0.47 ^a^	13.63 ± 0.99 ^b^
	*B. cinerea*	73.00 ± 1.16 ^d^	53.66 ± 1.91 ^c^	34.81 ± 2.0 ^b^	16.63 ± 0.88 ^a^
	*P. citrinum*	87.34 ± 2.22 ^d^	53.66 ± 2.06 ^c^	35.11 ± 1.50 ^b^	23.70 ± 2.22 ^a^

One-Way ANOVA, individual letters ^(a–d)^ in the upper case indicate the statistical differences between the concentrations; *p* ≤ 0.05; the negative values indicate a probacterial activity of the essential oil against the growth of microbial strains.

**Table 8 ijms-24-05179-t008:** Insecticidal activity of SSEO.

Concentration (%)	Number of Living Individuals	Number of Dead Individuals	Insecticidal Activity (%)
100	10	20	66.66
50	15	15	50.00
25	17	13	43.33
12.5	25	5	16.66
6.25	30	30	0.00
3.125	30	0	0.00
Control group	30	0	0.00

## Data Availability

Not applicable.

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
