# Peer review of "Salvia sclarea Essential Oil Chemical Composition and Biological Activities"

_ijms, 2023, doi:10.3390/ijms24065179_

Round 1
Reviewer 1 Report
This manuscript is very interesting to read and contributing greatly in the field of “Antimicrobials”. However, I have some minor comments to improve the manuscript.
Minor comments
Line 488 – 489: Authors should clarify the source of essential oil. Please clearly illustrate the source, whether extracted essential oil was used for GC-MS analysis or commercial source was used for the determination of compounds?
Please explain the following
Line 541: P. fluorescens was isolated from fish? Please explain the source (name of the fish), freshwater or marine? Isolated from the gut or other area? Confirmation test employed for biofilm production?
Line 553: Typo error, please correct
Line 556: nd samples were incubated at 37 °C for bacteria and 25 °C for 556
microscopic fungi in a duration of 24 hours. Inhibition zones were measured at thre
Plates were incubated at 37 °C, not samples please correct
In Tables 3, Table 4, 5, 7 authors inserted, Pseudomonas fluorescens biofilm. Please remove this row in all tables. This organism is only used for biofilm and antibiofilm studies? So collective illustrate separately and remove the row “Pseudomonas fluorescens biofilm” from antibacterial activity, MIC tables to avoid confusion. Other than this (biofilm), all are living organism. So, this variable is completely mismatch. Please check and do required changes throughout the manuscript.
Table 8: Inhibition zone value is not correlated with volume of oil. Authors performed triplicate experiments, so present the better results in tables.
Author Response
Reviewer #1
This manuscript is very interesting to read and contributing greatly in the field of “Antimicrobials”. However, I have some minor comments to improve the manuscript.
Response: First of all, we would like to thank the reviewer for the valuable suggestions. We have revised the manuscript according to the comments.
Minor comments
Point 1: Line 488 – 489: Authors should clarify the source of essential oil. Please clearly illustrate the source, whether extracted essential oil was used for GC-MS analysis or commercial source was used for the determination of compounds?
Response: We have reviewed the section in question. It is now: “Clary sage (Salvia sclarea) essential oil was purchased from Slovak company Hanus s.r.o. (Nitra, Slovakia). The essential oil was obtained by steam distillation of young branches and leaves of Salvia sclarea by the producer.”
Point 2: Please explain the following.
Line 541: P. fluorescens was isolated from fish? Please explain the source (name of the fish), freshwater or marine? Isolated from the gut or other area? Confirmation test employed for biofilm production?
Response: The information on P. fluorescens has been added in the material and methods part. Please see Line 549. As for the confirmation test, it was evaluated, and biofilm production was confirmed. P. fluorescens was previously confirmed by MALDI TOF MS Biotyper and sequencing. Those results were previously published please see the reference - https://doi.org/10.1016/j.micpath.2019.04.024.
Point 3: Line 553: Typo error, please correct.
Response: Accordingly, we have corrected the typo error.
Point 4: Line 556: nd samples were incubated at 37 °C for bacteria and 25 °C for microscopic fungi in a duration of 24 hours. Inhibition zones were measured at triplicate?
Plates were incubated at 37 °C, not samples please correct.
Response: We have introduced the correction according to the suggestion.
Point 5: In Tables 3, Table 4, 5, 7 authors inserted, Pseudomonas fluorescens biofilm. Please remove this row in all tables. This organism is only used for biofilm and antibiofilm studies? So collective illustrate separately and remove the row “Pseudomonas fluorescens biofilm” from antibacterial activity, MIC tables to avoid confusion. Other than this (biofilm), all are living organism. So, this variable is completely mismatch. Please check and do required changes throughout the manuscript.
Response: We have used the Pseudomonas fluorescens in biofilm testing (4.7), but also in the tests explained P. flourescens was tested in same way than other bacteria.
Point 6: Table 8: Inhibition zone value is not correlated with volume of oil. Authors performed triplicate experiments, so present the better results in tables.
Response: In this experiment, the inhibition zones are not correlated with the volume of oil, but it is correlated with the concentration of oil applied. The volume of essential oil was the same for all applied concentrations.
Reviewer 2 Report
ijms-2204765
Title: Salvia sclarea Essential Oil Chemical Composition and Biological Activities
Abstract: The abstract need to rewrite, because the authors write only the results of GC/MS analysis (the main constituents), but the results of antioxidants, antimicrobial, antibiofilm and insecticidal activity not represented.
Introduction: Prepared well.
Results:
2.3. Antimicrobial Activity in vitro
2.3.1. Disc diffusion method
Table 2. * Weak activity (zone 1–5 mm); ** Moderate activity (zone 5–10 mm); *** Strong activity (over 10 161 mm) according to what this evaluation.
Where is the statistical analysis of all data?
2.3.2. Minimal inhibitory concentration assay
Table 5, 6, and 8 need to be in one table under title MIC of SSEO, (E)-caryophyllene (Calculation of MIC50 and MIC90 need to revised because in all researches the results observed as MIC value only)
Table 7. Minimal inhibition concentration of meropenem in μg/mL. Why the authors determined the MIC of the antibiotic Meropenem.
Table 8. Minimal inhibition concentration of microscopic fungi in mm.
500 μL/mL of what?
There are several unclear points in the results. So, the results should be rewrite to observe the data well
Discussion: prepared well.
Materials and Methods:
4.5. Disc Diffusion Method
The reference of method, where is it?
4.6. Broth Microdilution Method
Where is the equation to calculate MIC50 and MIC90?
Conclusions: need to rewrite
There are several typing mistakes and grammatically errors. So, English editing is needed.
Author Response
Reviewer #2
Title: Salvia sclarea Essential Oil Chemical Composition and Biological Activities
Point 1: Abstract: The abstract need to rewrite, because the authors write only the results of GC/MS analysis (the main constituents), but the results of antioxidants, antimicrobial, antibiofilm and insecticidal activity not represented.
Response: We would like to thank the reviewer for all of the valuable suggestions. According to the suggestion, we have rewritten the abstract.
Point 2: Introduction: Prepared well.
Results:
2.3. Antimicrobial Activity in vitro
Point 3: 2.3.1. Disc diffusion method
Table 2. * Weak activity (zone 1–5 mm); ** Moderate activity (zone 5–10 mm); *** Strong activity (over 10 mm) according to what this evaluation.
Where is the statistical analysis of all data?
Response:
The activity was measured and evaluated by the experiences of our study with a comparison to other authors who evaluated zone size with the same method in different studies.
In Table 3, Table 4 and Table 5 it is not possible to do statistics because only one essential oil and one concentration was used and it is not possible to compare results between microorganisms because they do not belong to the same species and it is not possible to do a demonstration between them. The antimicrobial activity cannot be compared between different species of microorganisms. Table 5 shows the statistics because different concentrations within a single microorganism species are compared.
Point 4: 2.3.2. Minimal inhibitory concentration assay.
Table 5, 6, and 8 need to be in one table under title MIC of SSEO, (E)-caryophyllene (Calculation of MIC50 and MIC90 need to revised because in all researches the results observed as MIC value only)
Response: According to the suggestion we have merged all tables into one Table 5. in the manuscript. As for the MIC50 and MIC90 values, they are usually used as a representative, please see the following published manuscripts from other authors:
https://www.ncbi.nlm.nih.gov/pmc/articles/PMC3895893/
https://academic.oup.com/jac/article/57/3/498/738068
https://www.sciencedirect.com/science/article/pii/S1198743X14636442
https://www.jmbfs.org/issue/december-january-2018-19-vol-8-no-3/jmbfs_xxmc_0001/?issue_id=5070&article_id=20
Point 5: Table 7. Minimal inhibition concentration of meropenem in μg/mL. Why the authors determined the MIC of the antibiotic Meropenem.
Response: We have used meropenem as a positive control, to compare the efficiency of the tested EO and standard ((E)-caryophyllene) to the efficiency of the antibiotic known for its broad antibacterial coverage.
Point 6: Table 8. Minimal inhibition concentration of microscopic fungi in mm.
500 μL/mL of what?
Response: We have revised Table 8. according to suggestions, and we hope is clearer now.
Point 7: There are several unclear points in the results. So, the results should be rewrite to observe the data well
Response: We have revised the whole manuscript according to your comment.
Discussion: prepared well.
Materials and Methods:
Point 8: 4.5. Disc Diffusion Method
The reference of method, where is it?
Response: We have added the reference according to the suggestion.
Point 8: 4.6. Broth Microdilution Method
Where is the equation to calculate MIC50 and MIC90?
Response: The MIC50 and MIC90 values are calculated using the probit analysis, please see part 4.9. That is the usually used method, used in several previously published papers, please see the references below:
https://www.ncbi.nlm.nih.gov/pmc/articles/PMC3895893/
https://academic.oup.com/jac/article/57/3/498/738068
https://www.sciencedirect.com/science/article/pii/S1198743X14636442
https://www.jmbfs.org/issue/december-january-2018-19-vol-8-no-3/jmbfs_xxmc_0001/?issue_id=5070&article_id=20
Point 8: Conclusions: need to rewrite.
Response: We have revised the conclusion part of the manuscript according to your comment.
Point 9: There are several typing mistakes and grammatically errors. So, English editing is needed.
Response: We have revised the conclusion part of the manuscript according to your comment.
Reviewer 3 Report
The proposed manuscript “Salvia sclarea Essential Oil Chemical Composition and Biological Activities” is based on reasonable idea with scientific and practical importance, the experiment is well designed and performed, the manuscript is well composed and written, and the results are well presented. There are, though, parts, that could be improved/rewritten. The overall impression is for a not focused enough study.
Therefore I could suggest major revision to be done before accepting for publication, namely:
It is not clear enough why the antimicrobial activity of E-caryophyllene, a constituent in relatively low concentration (5.1 %) was chosen for comparisson with the standard antibiotic meropenem, instead of major components linalyl acetate (49.1%) and linalool (20.6%), both known for numerous biological activities, incl. antimicrobial. Please, add explanation.
I would recommend the representative GC/MS (TIC) chromatogram of the studied sample to be included as a Figure.
In Table 1, the components are listed according to their elution order on the HP-5MS column, though the component No 13 (a-Terpinene) with the RI 1016 is listed after the component with the RI 1047. Please, reconsider the identification of the compound. It is not clear, whether the RI values, presented in the table are measured experimentally or are based on the literature data (should be included both). All the “a-“, “b-“, etc. in the compound names should be in Symbol font.

Author Response
Reviewer #3
The proposed manuscript “Salvia sclarea Essential Oil Chemical Composition and Biological Activities” is based on reasonable idea with scientific and practical importance, the experiment is well designed and performed, the manuscript is well composed and written, and the results are well presented. There are, though, parts, that could be improved/rewritten. The overall impression is for a not focused enough study.
Therefore I could suggest major revision to be done before accepting for publication, namely:
Point 1: It is not clear enough why the antimicrobial activity of E-caryophyllene, a constituent in relatively low concentration (5.1 %) was chosen for comparison with the standard antibiotic meropenem, instead of major components linalyl acetate (49.1%) and linalool (20.6%), both known for numerous biological activities, incl. antimicrobial. Please, add explanation.
Response: We have used meropenem as a positive control, to compare the efficiency of the tested EO and standard ((E)-caryophyllene) to the efficiency of the antibiotic known for its broad antibacterial coverage. (E)-caryophyllene was chosen because there are not a lot of previously published results on the activity of this compound compared to the linalyl acetate and linalool.
Point 2: I would recommend the representative GC/MS (TIC) chromatogram of the studied sample to be included as a Figure.
Response: According to your suggestion, we have added a representative GC/MS chromatogram as supplementary Figure S1.
Point 3: In Table 1, the components are listed according to their elution order on the HP-5MS column, though the component No 13 (a-Terpinene) with the RI 1016 is listed after the component with the RI 1047. Please, reconsider the identification of the compound. It is not clear, whether the RI values, presented in the table are measured experimentally or are based on the literature data (should be included both). All the “a-“, “b-“, etc. in the compound names should be in Symbol font.
Response: According to your valuable comment, we have revised the whole Table 1. And introduced the changes. We have added in the table footer that presented RI values are calculated (obtained experimentally on HP-5MS column).
Round 2
Reviewer 2 Report
The authors corrected most of requirements
Author Response
Reviewer #2
The authors corrected most of requirements.
Response:
We would like to thank the reviewer for the comment.
Reviewer 3 Report
The authors have made corrections according to the reviewers’ recommendations and the manuscript has been substantially improved. It is still missing, though, an important information in Table 1, therefore, I would recommend minor revision to be done, before accepting for publication, namely:
The authors must provide literature data, as reported for DB-5 column (for example, according to NIST Chemistry WebBook; R.P. Adams EO Database or other sourse) for RI of all the compounds listed in Table 1, together with the experimentally measured in their study in order to support the identification.
Author Response
Reviewer #3
The authors have made corrections according to the reviewers’ recommendations and the manuscript has been substantially improved. It is still missing, though, an important information in Table 1, therefore, I would recommend minor revision to be done, before accepting for publication, namely:
Point 1: The authors must provide literature data, as reported for DB-5 column (for example, according to NIST Chemistry WebBook; R.P. Adams EO Database or other source) for RI of all the compounds listed in Table 1, together with the experimentally measured in their study in order to support the identification.
Response:
We would like to thank the reviewer for the valuable suggestion, accordingly, we have added RI (lit) in Table 1.